# Engineering and evaluation of FXa bypassing agents that restore hemostasis following Apixaban associated bleeding

Wojciech Jankowski [1,3], Stepan S. Surov [1,3], Nancy E. Hernandez [1], Atul Rawal [1], Marcos Battistel[2], Daron Freedberg [2], Mikhail V. Ovanesov[1] & Zuben E. Sauna [1] ✉

Direct oral anticoagulants (DOACs) targeting activated factor Xa (FXa) are used to prevent or treat thromboembolic disorders. DOACs reversibly bind to FXa and inhibit its enzymatic activity. However, DOAC treatment carries the risk of anticoagulant-associated bleeding. Currently, only one specific agent, andexanet alfa, is approved to reverse the anticoagulant effects of FXa-targeting DOACs (FXaDOACs) and control life-threatening bleeding. However, because of its mechanism of action, andexanet alfa requires a cumbersome dosing schedule, and its use is associated with the risk of thrombosis. Here, we present the computational design, engineering, and evaluation of FXa-variants that exhibit anticoagulation reversal activity in the presence of FXaDOACs. Our designs demonstrate low DOAC binding affinity, retain FXa-enzymatic activity and reduce the DOAC-associated bleeding by restoring hemostasis in mice treated with apixaban. Importantly, the FXaDOACs reversal agents we designed, unlike andexanet alfa, do not inhibit TFPI, and consequently, may have a safer thrombogenic profile.

Warfarin, a vitamin K antagonist, was the mainstay for managing thrombotic diseases over several decades[1]. The limitations of warfarin[1,2] led to the development of more specific, Direct Oral Anticoagulants (DOACs)[3]. Three commonly prescribed DOACs, rivaroxaban, apixaban, and edoxaban, are factor Xa (FXa) inhibitors[4,5] and have shown a favorable benefit-risk profile in the prevention and treatment of thrombotic events[6]. An important adverse effect associated with the use of these agents is acute major bleeding[7] that is difficult to treat. The annual risk of major bleeding is 2% to 4%, with fatality rates of 8% to 15%[8]. It has also been reported that most emergency department visits for medication harm were attributed to the therapeutic use of DOACs[9].

The only currently approved reversal agent for FXa inhibitors, andexanet alfa, is a catalytically inactive truncated variant of human FXa that binds and temporarily sequesters FXa inhibitors[10].

Andexanet alfa has several drawbacks: (i) andexanet alfa and FXa have comparable affinities for FXa inhibitors, explaining the unusually high (either 880 mg or 1,760 mg) dose (and cost) of treatment[11]. (ii) Recent meta-analysis of reversal agents demonstrated that effective hemostasis rate using andexanet alfa is comparable with Prothrombin Complex Concentrates (PCCs) that are used for the nonspecific reversal of DOACs[12,13]. (iii) The high doses of andexanet alfa have an off-target procoagulant effect by suppressing the activity of the tissue factor pathway inhibitor (TFPI), resulting in arterial and venous thrombosis[14]. An alternative strategy for reversing bleeding associated with the use of FXa inhibitors is the engineering of FXa variants that do not bind to the DOACs and could be used at catalytic concentration to stimulate hemostasis even in the presence of DOAC[15]. Two such agents have been previously designed[16,17].

[1]Hemostasis Branch 1, Division of Hemostasis, Office of Plasma Protein Therapeutics, Office of Therapeutic Products, Center for Biologics Evaluation & Research, US FDA, Silver Spring, MD, USA. [2]Laboratory of Bacterial Polysaccharides, Division of Bacterial, Parasitic and Allergenic Products, Office of Vaccines Research and Review, Center for Biologics Evaluation & Research, US FDA, Silver Spring, MD, USA. [3]These authors contributed equally: Wojciech Jankowski, Stepan S. Surov. ✉e-mail: Zuben.Sauna@fda.hhs.gov

Computational structural biology has considerable potential in the design and engineering of biomolecules with desirable interactions[18]. The approach involves testing and ranking thousands of variants in silico, allowing the selection of a limited subset of designs for in vitro screening. Final candidates can be comprehensively evaluated both in vitro and in vivo for safety and efficacy.

Here, we used two different design strategies that relied on two independent computational approaches. The first approach relied on RosettaDesign[19], whereas the second approach used the DALI[20] server that allows the identification of close structural matches. We demonstrate that two variants of FXa showed procoagulant activity in the presence of apixaban. We selected apixaban as the model as it is the most prescribed DOAC[21] and is forecasted to be the fourth-highest seller among all pharmaceuticals in 2024[22]. The novel FXa variants we designed could reverse apixaban-induced bleeding in a mouse model. Importantly, these reversal agents, unlike andexanet alfa, do not inhibit TFPI, which is associated with increased thrombogenicity, therefore demonstrating a potentially more favorable safety profile.

## Results

### Design of FXa variants with reduced DOACs sensitivity

FXa is a vitamin K-dependent 488 amino acids serine protease containing a heavy chain and a light chain that plays a critical role in the formation of the prothrombinase complex and is a target of DOACs due to its role in both the intrinsic and extrinsic coagulation pathways[23,24]. The DOACs bind to the active site of FXa and block the access to its substrate. Structural analysis of FXa in complex with apixaban shows that it interacts with amino acids located on subsites S1 and S4 of the active site[25]. S1 is mainly composed of G216, A190, C191, Q192, D189 and Y228 (Fig. 1a, b, highlighted in orange), and S4 is defined by the side chains of Y99, F174, and W215 (Fig. 1a, b, highlighted in green)[26]. Apixaban binds into the S1 subsite of FXa in a highly complementary manner via a pyrazole N-2 nitrogen that interacts with

the backbone of N192 and a carbonyl oxygen that interacts with the G216. The S4 subsite forms a narrow hydrophobic pocket (Fig. 1a, b, green), creating optimal space for the P4 ring of apixaban (Fig. 1a-b, green) which allows for high selectivity.

The objective of our computational strategies was to design novel FXa variants that are catalytically active in the presence of the DOAC apixaban. The first computational design strategy was based on a RosettaDesign protocol utilizing a FXa structure without apixaban (see "Methods" section). The process involved two rounds of design optimization: (i) Using the Rosetta Energy function (see Methods and Supplementary XML files) as a metric to estimate protein stability upon amino acid substitutions, (ii) Introduction of mutations to decrease the favorable interactions between FXa and apixaban but avoid sites on the protein identified in (i) with unfavorable energies. The goal of this process was to minimize protein destabilization due to aggressive mutagenesis while maintaining the FXa activity and preventing apixaban binding. All designs included mutations in an 10 Å region around the binding site of apixaban but excluded mutations within the catalytic triad necessary for FXa activity (Fig. 1c, **left**). The top-scoring (total_score) designs were then threaded with the FastRelax protocol onto the FXa-apixaban complex structure to construct final models (see "Methods" and provided res files for mutations). A manual inspection of 24 top-scoring models was carried out to determine whether the mutations could be accommodated in the binding pocket in the presence of apixaban. This involved identifying clashes with apixaban, any novel interactions, such as hydrogen bonding, favorable hydrophobic filling, and electrostatics, within the protein cavity. Following manual inspection and experimental evaluation/screening of selected variants, we made the final determination of mutations that provide the optimal trade-off between reduced apixaban binding and adequate clotting activity. This resulted in 9 designs for further evaluation.

For the second computational strategy, we used a homology-based design. This approach does not involve inserting mutations but

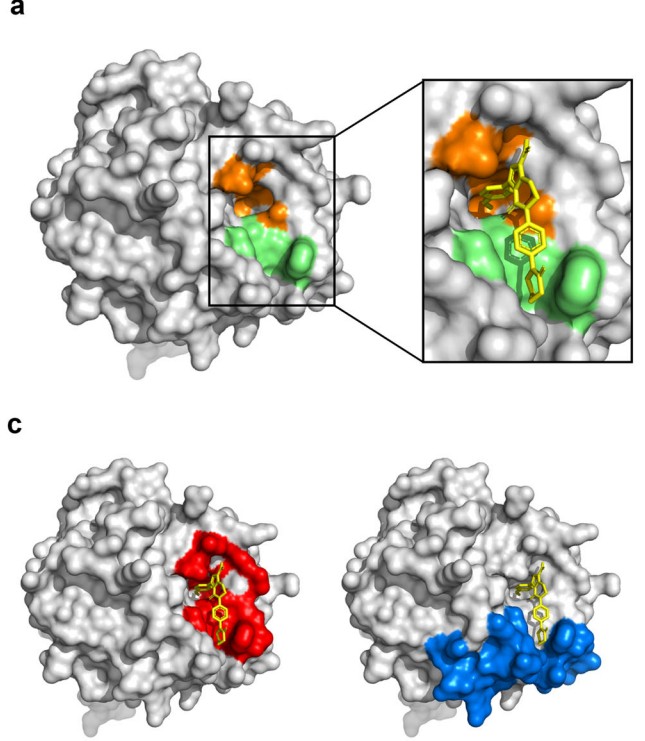

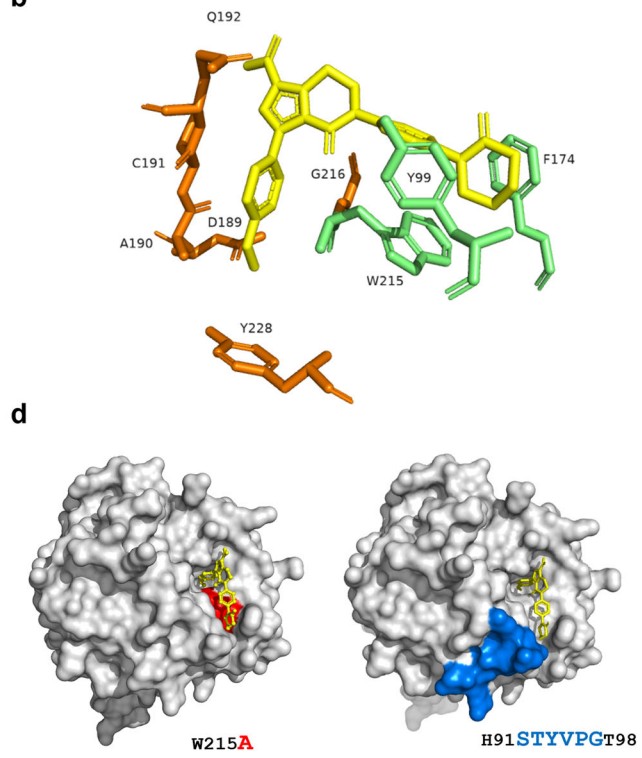

**Fig. 1 | Structural features of Factor Xa. a** Catalytic domain of FXa with highlighted S1 subsite (orange) and S4 subsite (green), and close-up view of apixaban (yellow) FXa complex. **b** Stick representation of apixaban orientation in the context of the S1 and S4 subsites. **c** Selected regions using RosettaDesign (red) and homology insertion (blue). **d** Models of FXa variants RDR2_2 (red) and HI_8 (blue) showing the amino acid substitutions.

rather swapping in segments of a homologous protein. To identify homologous proteins, we used the DALI server. This computational tool is used for comparing protein structures using a distance-matrix alignment. A statistical measure of the deviation from the mean (Z score) was used to identify structural homology with the heavy chain of FXa. The top hits were vitamin K (VK)-dependent coagulation proteins (factor VII (FVII), prothrombin, protein C, and factor IX (FIX)). These proteins all exhibit a multidomain architecture and functional features that are similar to FXa. We evaluated FVII as a potential candidate protein for the homology swapping approach because of the same loop size as FX (Supplementary Fig. 1). Using nuclear magnetic resonance, we demonstrated strong binding of apixaban to FXa but could detect no measurable binding to FVIIa (Supplementary Fig. 2). The region of FVII we selected for swapping into FX was based on the proximity to the apixaban binding pocket and the catalytic site of FXa (Fig. 1c, right). Based on these criteria, a loop segment, STYVPG (GenBank: AAA51983.1; FVII amino acid positions 292 to 297), was used to replace a homologous site on FXa. The two computational design workflows resulted in nine Rosetta-based designs (RDR2_1-RDR2_9) and one homology insert-based design (HI_8) (Supplementary Table 1).

## Functional screening of the FXa designs

The 10 computationally designed FXa variants were screened using a substrate cleavage assay (Supplementary Fig. 3). The FXa variants RDR2_3, RDR2_5, and RDR2_8 exhibited no measurable activity. We evaluated apixaban-mediated inhibition of FXa activity in the remaining 7 FXa variants (Supplementary Fig. 4). Among the FXa variants designed using the RosettaDesign protocol, RDR2_2 showed the largest increase (>350-fold) in the $IC_{50}$ for apixaban. Similarly, the homology-insert based design (HI_8) showed ~70-fold increase in the $IC_{50}$ value. We selected RDR2_2 and HI_8 FXa for comprehensive in vitro and in vivo characterization. Structural models of the variants RDR2_2 and HI_8 highlighting the amino acid substitutions are depicted in Fig. 1d (RDR2_2 highlighted in red and HI_8 highlighted in blue).

## Procoagulant potential of FXa variants RDR2_2 and HI_8

Although the FXa variants RDR2_2 and HI_8 show catalytic activity in chromogenic substrate cleavage assay, the activity is lower than that observed for wild-type FXa (wt-FXa) (Fig. 2a). The reduced activity can be explained by the disruption of the apixaban binding pocket which is necessary to disrupt apixaban binding. This hypothesis was supported by Molecular Dynamics (MD) simulations, which demonstrate that compared to wt-FXa there is an increase in the volume of the apixaban binding pocket of RDR2_2. The increase in the volume of the apixaban binding pocket also possibly translates to a loss of contact with P4

apixaban moiety and, consequently, lower sensitivity towards FXa inhibitors. HI_8, on the other hand shows a decrease in the volume of the apixaban binding pocket compared to the wt-FXa. Notable movement of the HI_8 loop region is observed in all four simulation runs when compared to the wt-FXa, which had a more stable root-mean-square deviation (RMSD) throughout the 600 ns simulation period. This suggests steric hindrance between the flexible region and possible apixaban (or any drug), further impeding the binding between the drug and the protein. The results of the MD simulations are depicted in Supplementary Figs. 5–8 and show that apixaban binding is obstructed in both RDR2_2 and HI-8 but that different molecular mechanisms may be involved in the two variants.

In addition to the activated FX we also evaluated the effect of introducing mutations on the FX zymogen/proenzyme. We measured both the intrinsic (using the aPTT assay) and extrinsic (using the PT assay) activities of the FX zymogen. Compared to wild type FX, the activity of zymogen RDR2_2 variant was reduced to 2% and 7% as measured by the aPTT and PT assays respectively. The zymogen HI_8 variant showed 0.5% residual PT activity and no measurable activity in the aPTT assay (Fig. 2b). This data suggests that administration of FX in the activated form (FXa) may be necessary for the reversal of DOAC associated bleeding.

Having established that the FXa variants are functional we evaluated the activity in the presence of increasing concentrations of apixaban. RDR2_2 and HI_8 exhibited ~350- and ~70-fold higher $IC_{50}$ values respectively for apixaban compared to wt-FXa (Fig. 2c).

## RDR2_2 and HI_8 activity measured using the Thrombin Generation Assay in apixaban and rivaroxaban-treated plasma

We carried out experiments in the physiologically relevant normal pooled plasma (NPP) instead of media by spiking NPP with FXa DOACs apixaban and rivaroxaban at low (100 nM) and high (2 μM) concentrations. The low concentration corresponds to the recommended clinical doses of 2.5 and 10 mg for apixaban and rivaroxaban respectively[27]. In this proof-of-concept study, we have used, as a worst-case scenario, a 20-fold higher concentration of the DOACs in the preclinical (murine) studies (see next section). Consequently, we included this higher concentration of DOACs in our in vitro evaluation. The gray bands in Fig. 3 represent FXa activity in NPP in the absence of DOACs. Addition of both low (Fig. 3a, c) and high (Fig. 3b, d) concentrations of apixaban (Fig. 3a, b) or rivaroxaban (Fig. 3c, d) results in loss of FXa activity. The FXa activity can be restored to physiological levels (gray band) in a dose-dependent manner by the addition of either RDR2_2 or HI_8. In all assays, considerably lower levels of FXa-variant RDR2_2 were required to restore FXa activity to physiological

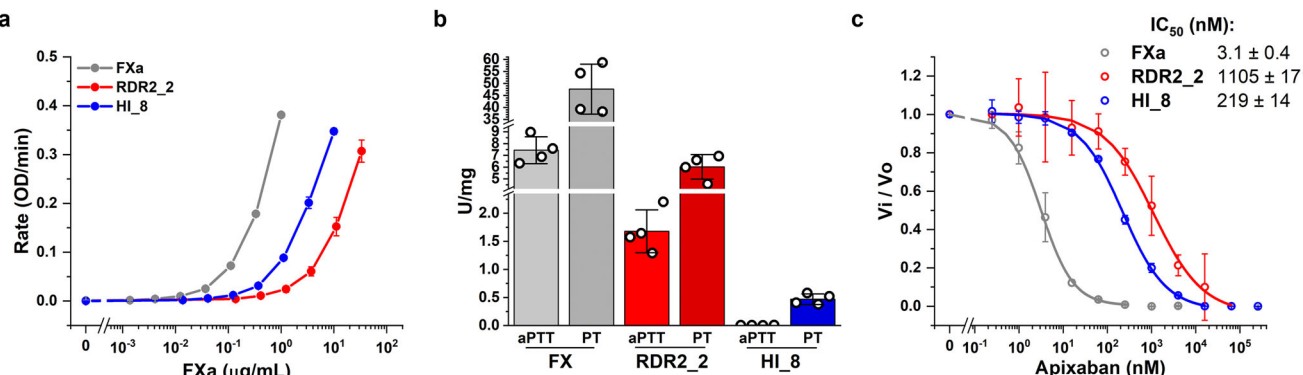

**Fig. 2 | In vitro characterization of FXa variants. a** The rate of chromogenic substrate cleavage by increasing concentrations of wt-FXa and variants in buffer system. FXa (Prolytix), a commercially available plasma derived FXa, was used to evaluate in-house made FXa to determine if activities are comparable (n = 3). **b** Zymogen specific clotting activity of wt-FX, and variants, RDR2_2 and HI_8 in aPTT- and PT-based assays (n = 4). **c** The rate of chromogenic substrate cleavage by 0.5 μg of FXa variants in the absence (Vo) or presence (Vi) of increasing apixaban concentrations. Values for $IC_{50} \pm SD$ were obtained from fitted curves (n = 3). All data represent mean ± SD. FXa/FX, RDR2_2 and HI_8 are shown in gray, red, and blue, respectively.

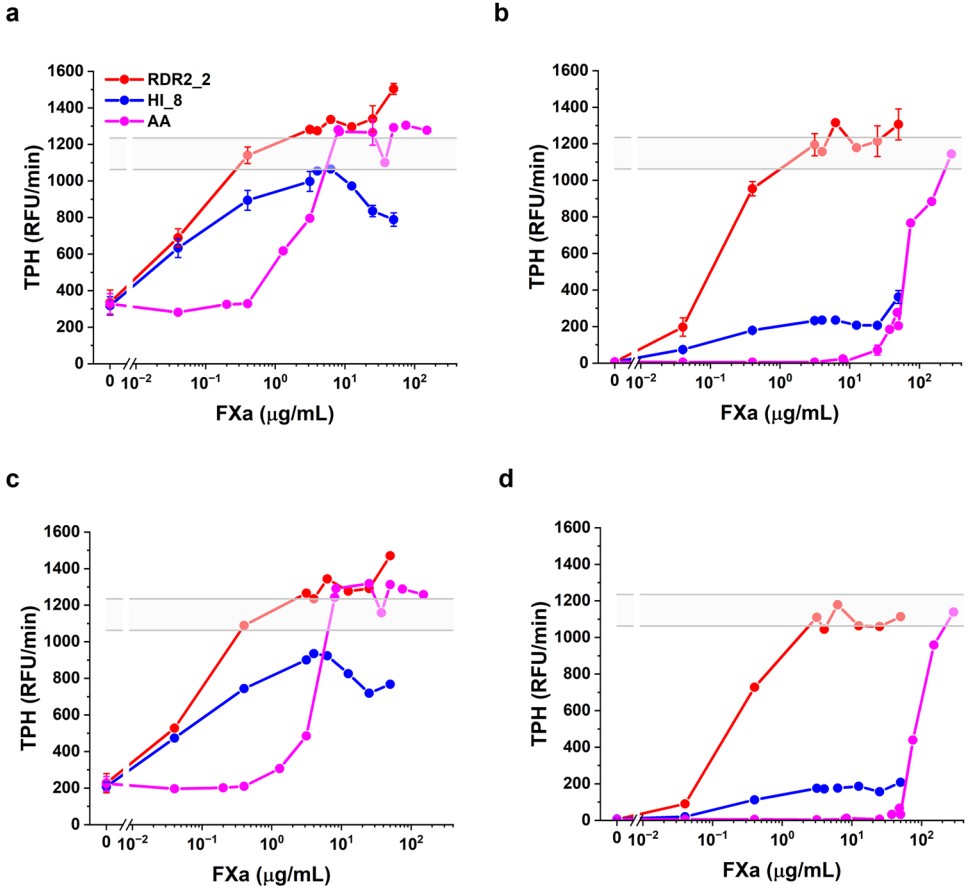

**Fig. 3 | Thrombin generation assay in the presence of inhibitor.** Effect of RDR2_2 (red), HI_8 (blue), and andexanet alfa (AA; magenta) on thrombin generation in the presence of apixaban (**a**, **b**) and rivaroxaban (**c**, **d**) treated plasma. Normal human plasma was spiked with 100 nM of apixaban (**a**) and 100 nM of rivaroxaban (**c**) or 2000 nM of apixaban (**b**) and 2000 nM of rivaroxaban (**d**) with 2 pM tissue factor and 4 μM phospholipids. Samples were treated with different concentrations of RDR2_2, HI_8 or andexanet alfa. The gray horizontal area represents the average (± SD) thrombin peak obtained for untreated normal pooled plasma. Each experiment was repeated 4 (for rivaroxaban) or 5 (for apixaban) times. The identical concentrations of FXa were not used in all runs. Data represent mean ± SD (bars showing SD are not depicted if a particular concentration was not measured in at least three runs).

levels, and this effect was accentuated at the higher (2 μM) concentrations of the DOACs. This experiment also compared RDR2_2 and HI_8 to andexanet alfa. In the presence of 100 nM apixaban/rivaroxaban FXa activity was restored by both RDR2_2 and HI_8 at concentrations ~20 fold lower than andexanet alfa.

### RDR2_2 and HI_8 restore hemostasis in mice treated with apixaban

We selected a mouse tail bleeding model, demonstrated to be sensitive to the anticoagulant effects of apixaban[10,16,28] to evaluate the in vivo reversal of apixaban associated bleeding (Fig. 4a). Based on two parameters (bleeding time and blood loss) we determined that an apixaban dose of 4 mg/kg was sufficient to induce bleeding (Supplementary Fig. 9). Mice treated with apixaban showed a significant increase in both bleeding time and blood loss compared to controls. The addition of either the RDR2_2 or HI_8 variants of FXa resulted in a significant decrease in bleeding time (Fig. 4b) and blood loss (Fig. 4c). Similar reversal of bleeding was not observed following the addition of FXa (1 mg/kg) or andexanet (5 or 25 mg/kg) (Fig. 4b, c). The larger dose of andexanet alfa is approximately equivalent to the high-dose clinical regimen of 1760 mg. Our in vitro results (Fig. 3) show that andexanet alfa is ineffective in reversing bleeding induced by high doses of apixaban, such as those used in the animal study.

Bleeding graphs demonstrated that hemostasis mediated by RDR2_2 and HI_8 prevents blood loss as evidenced by multiple breaks in blood flowing from the tail cut (Fig. 4d). Conversely, animals treated with apixaban (with or without andexanet alfa or wt-FXa) experienced mostly uninterrupted flow of blood (Fig. 4d).

### Thrombogenic potential of RDR2_2 and HI_8

To evaluate the thrombogenic potential of RDR2_2 and HI_8 we used the TFPI activity assay that measures the TFPI-mediated inhibition of the catalytic activity of the TF/FVIIa complex. Consistent with previous reports[29], andexanet alfa showed strong inhibition of TFPI activity (Fig. 5). However, even at high concentrations, both RDR2_2 and HI_8 show negligible inhibition of TFPI (Fig. 5). Thus, the FXa variants designed by us to circumvent DOAC associated bleeding, do not inhibit TFPI. These data indicate that TFPI-mediated thrombogenic risk may be lower for RDR2_2 and HI_8 compared to andexanet alfa (which inhibits TFPI activity).

## Discussion

The use of DOACs that target FXa is rapidly expanding[2], but a small percent of patients treated with DOACs show acute major bleeding, which can be fatal[30]. Although severe bleeding occurs in a small percentage of patients, DOAC-associated bleeds are an important clinical concern due to the large (and growing) number of individuals treated with DOACs. A rapid reversal agent specific to DOACs, which is effective and safe, remains an unmet medical need. Two products currently used to control bleeding-associated FXa-targeting DOACs

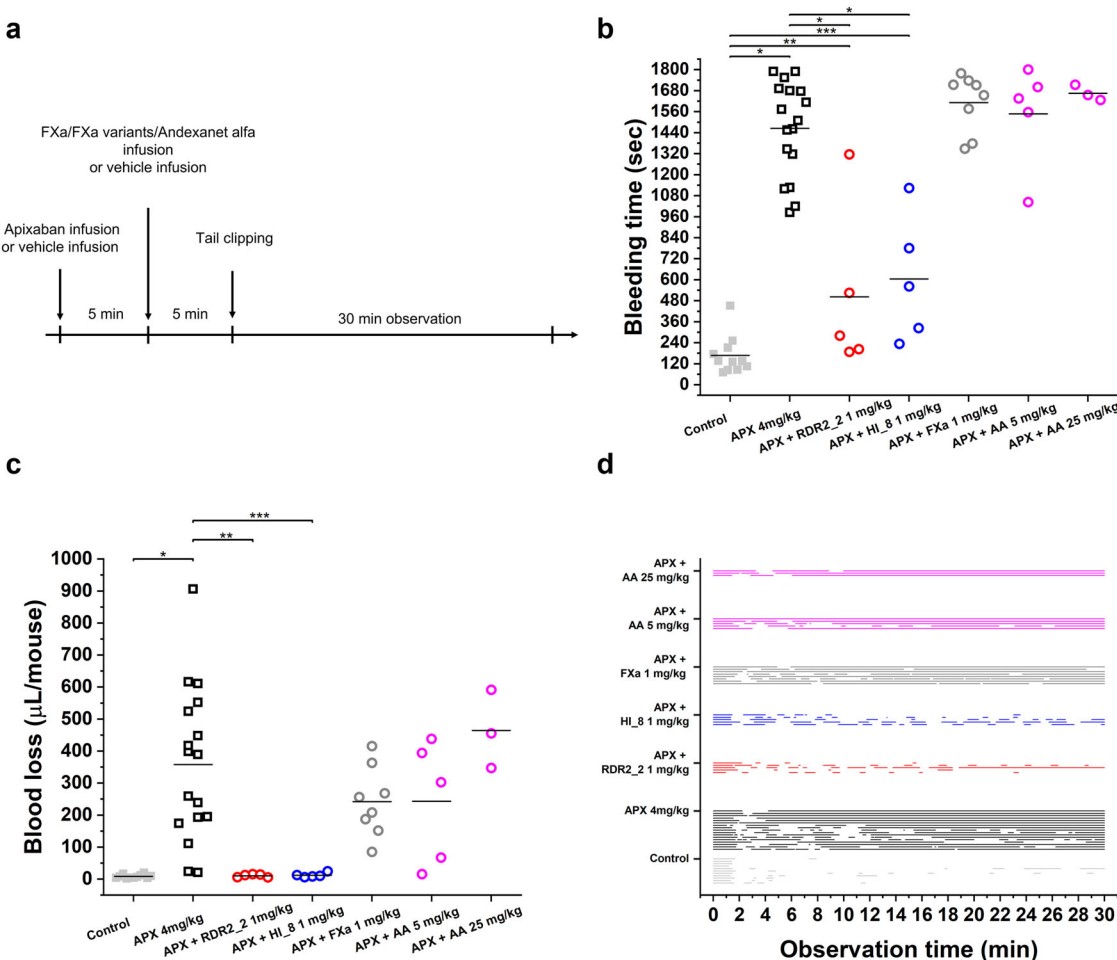

**Fig. 4 | Effect of FXa, RDR2_2, HI_8, and andexanet alfa (AA) on in vivo bleeding in the presence of apixaban (APX). a** Experimental design of the in vivo study. Three parameters were evaluated in the mouse tail-clipping model: **b** bleeding time; **c** blood loss; and **d** individual bleeding profiles in treated mice, where each line represents the bleeding episode in a single mouse. Variants RDR2_2 and HI_8, unlike wild type FXa and andexanet alfa, improved bleeding time in apixaban treated animals (* $P < 0.0001$, ** $P = 0.0375$, *** $P = 0.00199$, two-sided t-test), (**b**). Compared to animals treated with APX alone, treatment with variants RDR2_2 and

HI_8 resulted in reduced blood loss observed in control animals not treated with APX (* $P < 0.0001$, ** $P = 0.00447$, *** $P = 0.00463$, two-sided t-test) (**c**). Similar improvements in bleeding time and blood loss were not observed in animals injected with FXa or andexanet alfa. The number of animals for each of the treatment groups depicted in the figure is as follows: Control ($n = 11$), APX 4 mg/kg ($n = 17$), APX + RDR2_2 1 mg/kg ($n = 5$), APX + HI_8 1 mg/kg ($n = 5$), APX + FXa 1 mg/kg ($n = 8$), APX + AA 5 mg/kg ($n = 5$), APX + AA 25 mg/kg ($n = 3$). Black horizontal lines (**b**, **c**) indicate means within each group.

are (i) andexanet alfa (a catalytically inactive variant of human FXa which temporarily sequesters FXa inhibitors[10,13] is the only approved specific antidote for FXaDOACs and (ii) PCCs[31] (a nonspecific product that has not been approved for reversal of the direct FXa inhibitors). The efficacy of these strategies has not been demonstrated in well-controlled double-blinded clinical trials and their use is associated with about 10% risks of thrombotic adverse events within 30 days after treatment[14]. The increased risks of thrombosis may be due to the inhibition of TFPI in the case of andexanet alfa and an increase in plasma levels of prothrombin, FX and FIX following use of PCCs. However, it remains to be demonstrated if the risk is associated with the reversal strategy itself, the lack of anticoagulation or underlying risks in the patients.

An alternate strategy for the reversal of bleeding caused by DOACs is to engineer FXa variants that are insensitive to inhibition by DOACs. Here we report the design and characterization of FXa variants using two independent structure-based computational approaches. The design process included: (i) Design of substrate binding site which is also the DOACs binding pocket. (ii) Identification of mutations that could be deleterious vis-à-vis the stability or enzymatic function of the FXa molecule. (iii) Insertion of sequences of human origin and/or

minimizing the number of amino acid substitutions to minimize the risk of immune responses to the engineered protein (immunogenicity). (iv) Disruption of the apixaban binding pocket. The primary advantages of starting with a computational approach are high throughput and a very large design space. Given the current state of the art, despite the use of sophisticated computational design tools and principles, there is the possibility that the protein designed fails to meet one or more of the critical criteria (stability, FXa enzymatic function, lack of inhibition by a DOAC).

To enhance our chances of success, we adopted two completely different computation strategies. In the first, we used the RosettaDesign protocol to introduce mutations in the apixaban binding site of FXa, while in the second approach, we sought to insert a segment of a homologous human protein; all designs were engineered to be insensitive to apixaban. The computational approaches resulted in 10 FXa variants that were expressed, purified, and screened, first for FXa activity and then for decrease in the $IC_{50}$ for apixaban. Two variants, RDR2_2 and HI_8 were extensively characterized as potential bypassing agents for the reversal of DOAC associated bleeding.

In the presence of apixaban, both variants showed significantly higher activity (Fig. 2c) than wild type. The $IC_{50}$ (apixaban) values for

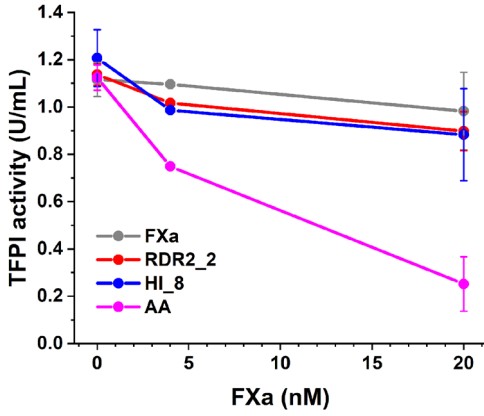

**Fig. 5 | Assessment of thrombogenic potential.** TFPI activity measurements following spiking of FXa variants (wt-FXa – gray, RDR2_2 – red, HI_8 – blue, andexanet alfa (AA) – magenta) into human normal pooled plasma at designated concentrations. $n = 2$-3, data represent mean ± SD (SDs are not shown for concentrations measured less than three times).

both RDR2_2 and HI_8 are orders of magnitude higher than the IC$_{50}$ (apixaban) for wt-FXa.

The results of an in vitro experiment conducted in NPP show that the addition of apixaban or rivaroxaban abolished the FXa activity in NPP, which could be reversed using RDR2_2 or HI_8 (Fig. 3a & c). Reversal of FXa inhibition was attained using far lower concentrations of RDR2_2 or HI_8 compared to andexanet alfa. This observation is consistent with the different mechanisms by which andexanet alfa and the variants designed by us reverse the anticoagulant effects of DOAC. Andexanet alfa, a non-functional FXa enzyme mutant binds to and sequesters the DOAC. As andexanet alfa and wt-FXa have comparable affinities, high concentrations of andexanet alfa are required to reduce the concentration of free DOAC. RDR2_2 and HI_8, on the other hand, are functional even in the presence of the DOACs (Fig. 2c) and can reverse FXa activity at much lower (catalytic) concentrations. It is important to note that at very high concentrations (2000 nM) of apixaban and rivaroxaban RDR2_2 is the only molecule that can reverse the inhibition of FXa activity (Fig. 3b & d). The high concentrations of DOACs are relevant because it was necessary to utilize this concentration to demonstrate consistent reversal of DOAC mediated bleeding in an in vivo animal model.

To convincingly demonstrate reversal of DOAC mediated bleeding it is first essential to identify the concentration of the anticoagulant that will consistently induce bleeding, which was determined to be 4 mg/mL of apixaban (Supplementary Fig. 9). Both FXa variants, RDR2_2 and HI_8, reversed the bleeding induced by apixaban. No reversal of bleeding was observed by using wt-FXa. At the concentration of apixaban used, the currently approved reversal agent, andexanet alfa, also did not reverse bleeding even though the concentration of andexanet was >6-fold higher than that used for RDR2_2 and HI_8. This observation is consistent with our findings in the in vitro studies showing that the presence of high concentrations of apixaban, andexanet alfa does not reverse the DOAC-induced inhibition of FXa activity (Fig. 3b).

We demonstrate here that, in vitro, RDR2_2 and HI_8 can reverse the inhibition of FXa activity by apixaban and rivaroxaban at concentrations that are considerably lower than those required for andexanet alfa. In a mouse model, a concentration of 1 mg/kg was sufficient to reverse bleeding induced by 4 mg/kg of apixaban. The consensus with respect to translating the animal dose to a human equivalent dose is that rather than a conversion based on relative body weights, a body surface area normalization method correlates well with physiological and clinical outcomes[32,33]. Based on this method, the

projected clinical dose of FXa variants RDR2_2 and HI_8 translates to approximately 5.7 mg for a 70 kg adult. Note that by this same method, the dose of apixaban in mice that is equivalent to the human dose is 0.44 mg/kg while we used an ~10-fold higher dose of 4 mg/kg. The projected dose for FXa variants is thus at the high end of the therapeutic window and can be titrated down. Nonetheless, even the high end of the projected dose of RDR2_2 or HI_8 (5.7 mg for a 70 kg adult) is considerably lower than the doses of andexanet alfa (880 and 1760 mg for an adult).

Although our data suggests that RDR2_2 and HI_8 could be attractive drug candidates for reversing apixaban (and plausibly rivaroxaban) induced bleeding, potential risks remain. Like all biologics, both variants could elicit immune responses in humans[34]. However, only a single mutation was introduced in RDR2_2 while HI_8 was designed by swapping in another human sequence, and therefore, these designs are likely to have low immunogenicity risk. Another important risk associated with the currently used reversal agent andexanet alfa is prothrombotic events which are estimated to occur in ~10% of patients[12,13]. We compared TFPI activity in the presence of andexanet alfa, RDR2_2 and HI_8. While andexanet alfa showed strong, dose-dependent inhibition of TFPI activity, both RDR2_2 and HI_8 did not inhibit TFPI. Nonetheless, procoagulant activity of the variants we designed deserves further investigation because these molecules have proteolytic activity which andexanet alfa lacks. This potential procoagulant activity is evident as overcorrection of TG observed at high concentrations of the FXa variants (Fig. 3). The potential thrombogenic mechanism(s) differs from that of andexanet alfa because the variants we designed do not exhibit interactions with TFPI. It is nonetheless important to note that the observed overcorrection occurs at concentrations that are >10-times higher than the effective dose of the FXa variants (i.e., the concentration that is sufficient to restore hemostasis to normal level upon anti-FXa treatment).

Taken together our results indicate that two computational design strategies were successful in designing FXa variants with decreased affinity for apixaban and could effectively circumvent the DOAC-mediated inhibition of endogenous FXa. In a mouse model, these FXa mutants reversed bleeding associated with apixaban. Moreover, these new molecular entities are likely to be safer than existing treatments due to the likelihood of minimal immunogenicity and low TFPI-mediated thrombogenic potential.

## Methods

### Computational design

**RosettaFastRelax: Preparation of input structure.** The crystal structure of FXa (PDB ID:2P16 [https://www.rcsb.org/structure/2p16]) was used as the starting structure for an all-atom refinement process using the Rosetta Relax protocol. Two structures were generated. The first FXa structure did not include apixaban in the binding pocket. This structure was run through fixed backbone Rosetta energy minimization (FastRelax protocol, 1000 decoys generated which represent alternative conformations of the protein, see Supplementary Information) to identify favorable conformations of the side chains and the top-scoring (lowest energy) one was used for further design. The second design used FXa containing the apixaban inhibitor as the starting structure. The FastRelax protocol was then executed on this structure to investigate how the presence of apixaban affects the protein's structure and interactions with apixaban.

**RosettaDesign: Design of apixaban binding pocket.** The first round of designs used the RosettaDesign protocol (without fixed backbone) where all residues within 10 Å of apixaban, but excluding the catalytic triad necessary for FXa activity, were designated as the "cavity". The top-scoring delta "total_score" and delta "cavity_score" mutations were identified, and their sequence profiles were analyzed. The mutations identified through RosettaDesign were threaded with the FastRelax

protocol onto the structure of the FXa-apixaban complex (see Supplementary Information) to construct final models. Through manual inspection of the FXa variants, we either retained the mutation or reverted to the wild-type amino acid based on conventional structural biology principles. The resulting 21 designs were experimentally validated. Building on the experimental results from first round of design a second round of design was initiated. This round focused on creating conservative single- and double-point mutations that could hinder apixaban binding but retain functional activity. This second round of design resulted in 9 designs for further experimental validation. All script files used for design are provided in Supplementary Information as "*Computational scripts and files*" and files are provided in "zip file".

**Homology based designs.** DALI server[20] is a tool that uses a distance-matrix alignment method to compare protein structures and identify similarities. Using this tool, we identified structures similar to FXa in the human proteome, which could be introduced into the structure of FXa to prevent apixaban binding. The DALI server was provided with PDB ID 2P16 chain A and the top hits were filtered based on Z score (statistical measure of the deviation from the mean). FVII, another protein involved in the blood coagulation pathway, was identified as one of the top hits for structural similarity. The characteristics used to evaluate potential inserts included proximity to apixaban's binding pocket, proximity to FXa's catalytic site, length of insertion, and overall location. Based on these considerations, we inserted a loop segment extracted from FVII (STYVPG) onto a similar site on FXa.

**MD simulations.** Molecular Dynamics (MD) simulations were performed using GROMACS 2021.3 in conjunction with visualization software (VMD) and PyMOL[35–37]. Production simulations were performed for 600 ns under NPT conditions. RMSD for both the wild type-FXa (wt-FXa) and variant models were performed. This was done to investigate the deviation of the variant region over the dynamic 600 ns simulation. Pocket volume calculations for the binding pocket of the proteins were done via the Fpocket suite[38]. The pocket volume for the binding pocket was calculated for different timesteps of 600 ns simulations.

### Protein expression and purification
DNA constructs encoding FX variants were synthesized by Genscript (Piscataway, NJ, USA) and subcloned into pGeneLenti/eGFP expression vector. Lentivitral particles were prepared by Genscript and used to transduce HEK293T/17 cells (ATCC; CRL-11268). Stable cell lines expressing FX variants were cultured in FX-specific expression media[17]. FX variants were purified from conditioned media by CaptureSelect™ Factor X Affinity Matrix (Thermo Scientific) following the manufacturer's protocol. Eluted protein was concentrated and activated with RVV-X (Prolytix), followed by size exclusion chromatography (Superdex™ 75 Increase 10/300 GL).

### FXa amidolytic activity
The FXa activity assay was based on CS-11(65) chromogenic substrate (Hyphen BioMed). FXa samples with or without apixaban (Selleck Chemicals LLC) were diluted in Tris-BSA buffer pH 7.4 (Hyphen BioMed) containing 5 mM $CaCl_2$ and mixed with substrate (300 µM final concentration [f.c.]) at a ratio of 3:2, and absorbance (410 nm) was measured kinetically on a Biotek Synergy H4 (Agilent Technologies, Inc.) microplate reader at 37 °C.

### Specific activity
Specific activity of FX zymogens was measured using aPTT and PT clotting assays. In the aPTT assay, 1 part of the FX sample pre-diluted in Tris-BSA buffer was added to 1 part of FX-deficient plasma (Affinity Biologicals, Inc.), then mixed with one part of contact trigger and lipid reagent (SynthaSil, Instrumentation Laboratory) followed by 180 s incubation at 37 °C. Coagulation was initiated by the addition of one part of $CaCl_2$ to two parts of the reaction mixture, after which the clotting time was measured by monitoring clot absorbance at 671 nm on a BioTek Synergy Neo2 reader (Agilent Technologies, Inc.).

In the PT assay, FX sample and FX-deficient plasma were incubated for 60 s at 37 °C. Coagulation was triggered by the addition of two parts of PT reagent (RecombiPlasTin 2 G, Instrumentation Laboratory) to one part of the plasma mixture, and clotting time was measured in a microplate reader. Calibration curves consisted of FX with known activity (Enzyme Research Laboratories).

### Thrombin Generation Assay
The procoagulant activity of FXa variants (andexanet alfa was purchased from Creative Biomart Inc.) was evaluated using a previously described thrombin generation assay in NPP (Precision BioLogic Inc.)[31]. Samples were serially diluted in Tris-BSA buffer and mixed with a recalcified plasma mixture containing plasma (50% vol/vol f.c.), apixaban or rivaroxaban (0.1 µM or 2 µM f.c., both from Selleck Chemicals LLC), platelet substitute in the form of phosphatidyl choline, phosphatidyl serine and sphingomyelin phospholipid vesicles (4 µM f.c.; Phospholipid-TGT®, Rossix), recombinant lipidated TF (2 pM f.c.; RecombiPlasTin, Instrumentation Laboratory), fluorogenic substrate for thrombin ZGGR-AMC (800 µM f.c.; Bachem) and $CaCl_2$ (12.5 mM f.c.) with a robotic 96-channel pipettor (ViaFlo 96, Integra Biosciences). Fluorescence was measured at 37 °C using a Biotek Synergy H4 reader, and fluorescence curves were analyzed to obtain the Thrombin Peak Height (TPH) parameter, which was the most representative and responsible parameter of thrombin generation test. The normal TPH range was obtained for human plasma in the absence of inhibitors and FXa variants.

### Measurement of TFPI activity
TFPI activity was tested with a chromogenic assay (Actichrome TFPI, BioMedica Diagnostics) according to manufacturer's instructions with some modifications: samples were added to NPP in a 1:9 ratio (sample in buffer : NPP), then this mixture was diluted in TFPI depleted plasma (1:19).

### Tail clip bleeding model
Animal experiments were performed at the U.S. FDA/CBER Division of Veterinary Services in accordance with the protocol approved by the institutional animal care and use committee (IACUC). All mice in facility are maintained on a regular diurnal lighting cycle (12:12 light:dark) at a room temperature of 71 ± 1 °F and 30-70 % humidity, with ad libitum access to food (LabDiet 5P76 IsoPro RMH 3000 Irradiated) and automated water. Male CD-1 mice (Charles River Laboratories) aged 4-12 weeks were anesthetized with a mixture of ketamine and xylazine, placed on warming pad and administered apixaban (or control, DMSO). After 5 min FXa or NaCl (placebo control) were administered via retro-orbital injections[39]. After an additional 5 min, a 3 mm tip of the tail was cut, immersed in a tube with a predefined volume of 0.9% NaCl at 37 °C, and bleed data were collected for 30 min.

### Statistical analysis
All in vitro data are presented as mean ± standard deviation and are the result of at least three experiments unless otherwise stated. Plotting and data analysis of graphs was carried out using OriginPro software Version 10.0 from OriginLab.

### Reporting summary
Further information on research design is available in the Nature Portfolio Reporting Summary linked to this article.

## Data availability

All data are available within the Article, the Supplementary Files, or the Source data file that has been provided. Source data are provided in this paper.

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

## Acknowledgements

This study used the computational resources of the High-Performance Computing clusters at the Food and Drug Administration, Center for Devices and Radiological Health. This research was supported by funds from the Hemostasis Branch/Division of Plasma Protein Therapeutics/Office of Tissues and Advanced Therapies/Center for Biologics Evaluation and Research of the U.S. Food and Drug Administration. AR was supported by appointment to the Research Participation Program at the Center for Biologics Evaluation and Research administered by the Oak Ridge Institute for Science and Education through an interagency agreement between the U.S. Department of Energy and the U.S. Food and Drug Administration.

## Author contributions

W. J., S.S.S., M.V.O., N.E.H., and Z.E.S. conceived and designed the experiments; W. J., N.E.H., S.S.S, A.R., M.B. performed experiments; W. J., N.E.H., S.S.S, A.R., M.B., D.F., M.V.O., Z.E.S. analyzed the data W. J., N.E.H., S.S.S, A.R., M.B., D.F., M.V.O., Z.E.S contributed materials/analysis tools and W.J. and Z.E.S. wrote the paper.

## Competing interests

The authors declare no competing interests.
