## [Peer Review File · Nature Communications]

Engineering and evaluation of FXa bypassing agents that restore hemostasis following Apixaban associated bleedingREVIEWER COMMENTS

Reviewer #1 (Remarks to the Author):

I would like to congratulate the authors on their interesting work. This study outlines various strategies for designing factor Xa inhibitor bypassing agents. FXaI-associated bleeding is a significant clinical problem with wide reaching consequences. Existing specific reversal agents are costly, and availability is limited (e.g., andexanet alfa is not available in my country).

This work is important, but I do have some concerns with respect to the methodology used and conclusions re: hypercoagulability.

I would also encourage the authors revisit the introduction and discussion, and if possible, revise the wording/grammar/syntax. It might also be useful (if possible) to have a clinician (with expertise relating to anticoagulation reversal and anticoagulant-related bleeding) to review the wording and ensure the wording is accurate/valid from a clinical standpoint.

Please see below for my comments and suggestions. I hope these suggestions are helpful.

Comments to authors:

Abstract –

1. “only one specific agent is approved to reverse the anticoagulant effects of DOACs”. If you are referring to andexanet alfa, it might be better to specifically state this and specify “direct Xa inhibitors” or the acronym “FxaIs” throughout to avoid confusion. If you are referring to all DOACs, including dabigatran, than the statement above is incorrect (both idarucizumab for dabigatran reversal and andexanet alfa for FxaI reversal have been approved. Consider revisiting.
2. “However, because of its mechanism of action hundreds of milligrams are needed 28 for a therapeutic effect”. As a clinician, it does not matter to me whether the dosing is 400 mg or 2 mg. The dosage in milligrams is irrelevant to most clinicians. E.g., idarucizumab dosing is

5g total, but it is still used when needed. If you are trying to refer to andexanet alfa's cumbersome dosage schedule (bolus then infusion, different dosing dependent on the specific Fxal and timing since last dose), then I would explicitly state this. Consider revisiting.

3. "novel specific reversal variants that exhibit clotting". The phrasing "exhibit clotting" is clinically inaccurate. Clinicians/hematologists often refer to "clotting" in terms of pathological thrombosis (e.g., DVT, PE). Here I believe you are referring to the anticoagulation reversal activity of your novel specific variants. The way it is currently written, I am unsure whether you are referring to in vitro reversal of FXaI anticoagulant effect or hemostasis (i.e., cessation of bleeding) in an animal model. Consider revisiting.

4. "effective at low doses and do not inhibit TFPI, and consequently, are less likely to be thrombogenic". It's not the dose alone (e.g., mg) that would be relevant in terms of thrombogenicity. Affinity, mechanism, etc. would all affect thrombogenicity. If you are comparing to andexanet alfa, maybe molar equivalent, or binding affinity, would be a better way to compare relative amounts required to achieve a reversal effect.

5. Throughout the abstract you refer to a "specific agent" without naming the agent. Although this is implied after reading the entire article, this will be confusing to readers who view the abstract alone. If you are referring to andexanet alfa, then simply say andexanet alfa.

Introduction –

1. Typo - forecasted to be the fourth highest seller among all pharmaceuticals. Please revisit.

Results –

1. "Using NMR" – Please define acronyms with first use; if used only once, please spell out.

2. Same for RSMD, please define at first use.

3. “The increase in the volume of the apixaban binding pocket also possibly translates to loss of contacts with P4 apixaban moiety and consequently lower sensitivity towards FXa.”

Please revisit, do you mean “and consequently lower sensitivity towards FXa inhibitors”?

4. “The low concentration corresponds to the recommended clinical dose.” What clinical dose are you referring to (e.g., rivaroxaban 20 mg once daily, rivaroxaban 15 mg twice daily?). Please provide a reference to support this statement.

5. RDR2_2 and HI_8 activity in apixaban and rivaroxaban-treated plasma – Please refer to the fact that you are assessing activity using a thrombin generation assay. This was unclear to me on first readthrough until I saw the title of Figure 3.

6. “Addition of both low (Figs. 3A & C) and high (Figs. 3B & D) concentrations of apixaban (Figs. 163 3A & B) or rivaroxaban (Figs. 3C & D) result in loss of FXa activity.”. Are the figure labels you cite in this sentence correct? You state figures C and D are for apixaban, but in the Figure 3 legend, it says “apixaban (A, B) and rivaroxaban (C, D) treated plasma”. Please check. Consider adding descriptive labels for each letter A, B, C, D to make it easier to identify the data you are presenting.

7. Figure 3. In the introduction you propose that these Fxa variants are less thrombogenic than andexanet alfa due to lack of interaction/binding to TFPI. Your thrombin generation data seems to suggest an overcorrection of thrombin peak height (Figure 3A and C) at the low (clinically relevant) FXaI concentrations (100 nM). Could the RDR2_2 variant be leading to a hypercoagulable effect through a different mechanism?

8. “Thus, the FXa variants designed by us to circumvent DOAC associated bleeding, are unlikely to be thrombogenic.” ◇ Unlikely to inhibit TFPI. You cannot conclude the variants you have generated are unlikely to be thrombogenic based on this mechanism alone. Your TGA data shows evidence of hypercoagulability based on overcorrection of peak height, and in fact, the overcorrection parallels results obtained for andexanet alfa for the low (100 nM) FXaI concentrations (Figures 3A and C).

9. “This data suggests that administration of FX in the activated form (FXa) may be necessary for the reversal of DOAC associated bleeding.” What do you think the clinical implications of this might be? Could the administration of FXa lead to potential thrombotic risk? Could this be a potential explanation for the overcorrection of peak height seen in Figures 3A and C? Wouldn’t the administration of a zymogen be desirable since the bypassing protein would be regulated by endogenous coagulation processes? (e.g., VMX-C001 is a zymogen).

Discussion -

1. Typo – “Two products currently used to control bleeding associated FXa-targeting DOACs”. Please check.

2. “their use is associated with significant risks of thrombotic adverse events”. This statement is too vague. There is still debate (since it is hard to prove) that thromboembolic events post reversal are directly attributable to the reversal agent in question. Some of these thromboembolic events might just be due to the underlying indication for anticoagulation (e.g., atrial fibrillation and risk of stroke). I would consider rephrasing this and simply cite the 30-day risks of thromboembolic events observed post hemostatic therapy and mention that some of this risk might be attributable to the hemostatic therapy itself (e.g., TFPI interaction for andexanet alfa; administration of supraphysiological levels of prothrombin for PCCs). You do state “The increased risks of thrombosis may be due to inhibition of TFPI in the case of andexanet alfa and an increase in plasma levels of prothrombin, FX and FIX following use of PCCs.” later in the paragraph, but could also consider adding that some of these events may be due to underlying risk rather than the anticoagulation reversal itself.

3. “particularly when compared to an alternate reversal agent that includes sequences derived from the venom of the elapid snake (*Pseudonaja textilis*).”, a reference would be needed here, which reversal agent are you referring to?

Methods –

1. Re: thrombin generation assay, did you consider using a calibrated assay (calibrated to

thrombin units, rather than RFU). We're any other parameters collected (e.g., total thrombin generated; analogous to e.g., endogenous thrombin potential in the case of Calibrated Automated Thromography). If trying to estimate hypercoagulability, area under the thrombin generation curve might be a more appropriate measure than peak height. If both were measured, both should be reported, along with kinetic parameters (lag time and time to peak).

Reviewer #2 (Remarks to the Author):

The authors have engineered blood coagulation FX to produce antidote molecules that could be used by patients that receive direct oral anticoagulant molecules (DOACs) targeting factor Xa, such as apixaban and that would be at risk of bleeding, due to, for instance, a car accident or related threatening conditions. There is indeed a need for such "antidote" molecules as more people are likely to take anticoagulant therapies (eg, ageing of the population). The available agent that reverse the effects of FXa DOACs named andexanet alfa (catalytically inactive truncated variant of human FXa) has numerous adverse effects and drawbacks.

The study is therefore of high interest. Below are my questions/comments following the order of the article.

A) The authors wrote: The inhibitor binding site of FXa is located on the S1 and S4 sub-pockets of the catalytic domain

This tends to be confusing and could be optimised. The catalytic site of all serine proteases contain different subpockets, all pockets can contribute to the free energy of binding, some, in addition, are important for specificity/selectivity.

B) The authors want to design FXa variants that are catalytically active in the presence of the DOAC apixaban. They use a Rosetta protocol for this purpose, with two main goals:

B1) Identified mutations that would adversely affect FXa function or not permit efficient expression of the protein

This is unclear, to the best of my knowledge, Rosetta can not tell if a mutation can lead to

efficient expression of a protein? What seems to be done here is to probe if mutations around the active site would affect the stability of the protein? So it should be something like a $\Delta\Delta G$? The authors should explain in the supplement what is being computed and explain the Rosetta method used. The readers do not know if the provided scripts are commonly used in the field or if they carry something unique.

B2) The authors also want mutations that decrease the affinity for apixaban. Here it is unclear, the authors talk about top-scoring designs (with respect to binding site contact modifications)? Please translate. A predicted binding score? Creation of cavities/clashes among amino acids in this region and thus potential local destabilisation? What is being monitored? And then the authors investigate manually potential clashes between the mutant FXa and apixaban? Why not looking at binding scores directly? What if it is not a clash but modified electrostatic interactions? All this section is highly unclear and should be explained in the supplement.

B3) then the authors use the DALI server to identify other serine proteases from the coagulation that could give some insights to introduce mutations.

The reason for DALI is not clear here. It is well known that serine proteases from the coagulation are relatively similar, at least in the serine protease domain, with some key differences in amino acids here and there and in insertion/deletion of some loops. What would seem important, in addition of DALI or instead of DALI, is to have a multiple sequence and structural alignment that will show at the amino acid level, what is conserved, where we have insertion or deletion in the family so as to put a score on the importance of selecting some regions for mutations/grafting, to see which amino acids are risky to mutate while also showing the secondary structure elements of FXa... Such a multiple sequence alignment in the supplement would be welcome and is easy to do. The authors wrote that they measure apixaban binding to FVII using NMR. It is surprising that such data are not available for apixaban and most coagulation proteases (at least) as such information would be mandatory during the drug development process. As such, just taking this information would help to select some proteases from which to get some insights in term of mutation, and then go to the 3D. The selection of FVII STYVPG to replace a related loop in FX should be shown in the supplement (the loop has the same size... which could then give some insights

about potential immune reactions...).

What is known about the introduced mutations in FX? Are they present in patients?

C) Then the authors evaluate the procoagulant potential of the two selected FX variants.

They wrote: The reduced activity can be explained by the destabilization of the pocket

This is unclear, the pocket is not destabilised? The fitting of the inhibitor is perturbed, the pocket might become more flexible or more rigid... but destabilisation here would mean some kind of $\Delta\Delta G$ issue, and potentially folding issue.

D) In the abstract the authors talk about andexanet alfa, in the discussion they add PCCs, thus the two strategies should be mentioned in the introduction?

E) This sentence should be clarified: The design process included: (i) Design of an 'empty' apixaban binding pocket. What is an empty pocket?

F) The computer strategy is somewhat surprising as very simple alternative could lead to the same results (eg. Multiple sequence alignment and literature search).

In conclusion, the reported experimental data seem scientifically sound but several clarifications are missing with regard to the computational approaches. For instance, why the authors used fixed backbone Rosetta energy minimization while it is known that proteins are flexible and that small adjustments in the backbone are usually needed after introduction of amino acid changes? So again the computational approaches need to be explained further, if there is no room in the main paper, this can be done in the supplement. The way MD is used here is surprising, free energy of binding with the ligands could be computed for the wt and mutants instead or in addition of monitoring the pocket volume?

REVIEWER COMMENTS

Reviewer #1 (Remarks to the Author):

I would like to congratulate the authors on their interesting work. This study outlines various strategies for designing factor Xa inhibitor bypassing agents. FXaI-associated bleeding is a significant clinical problem with wide reaching consequences. Existing specific reversal agents are costly, and availability is limited (e.g., andexanet alfa is not available in my country).

This work is important, but I do have some concerns with respect to the methodology used and conclusions re: hypercoagulability.

I would also encourage the authors revisit the introduction and discussion, and if possible, revise the wording/grammar/syntax. It might also be useful (if possible) to have a clinician (with expertise relating to anticoagulation reversal and anticoagulant-related bleeding) to review the wording and ensure the wording is accurate/valid from a clinical standpoint.

Please see below for my comments and suggestions. I hope these suggestions are helpful.

Comments to authors:

Abstract –

1. “only one specific agent is approved to reverse the anticoagulant effects of DOACs”. If you are referring to andexanet alfa, it might be better to specifically state this and specify “direct Xa inhibitors” or the acronym “FxaIs” throughout to avoid confusion. If you are referring to all DOACs, including dabigatran, than the statement above is incorrect (both idarucizumab for dabigatran reversal and andexanet alfa for FxaI reversal have been approved. Consider revisiting.

Thank you for your suggestion. We clarified the sentence by changing “only one specific agent is approved to reverse the anticoagulant effects of DOACs” to “only one specific agent, andexanet alfa, is approved to reverse the anticoagulant effects of FXa-targeting DOACs (FXaDOACs)” (please see line 26-27). We have also used the acronym “FXaDOACs” where appropriate in the manuscript.

2. “However, because of its mechanism of action hundreds of milligrams are needed 28 for a therapeutic effect”. As a clinician, it does not matter to me whether the dosing is 400 mg or 2 mg. The dosage in milligrams is irrelevant to most clinicians. E.g., idarucizumab dosing is 5g total, but it is still used when needed. If you are trying to refer to andexanet alfa’s cumbersome dosage schedule (bolus then infusion, different dosing dependent on the specific FxaI and timing since last dose), then I would explicitly state this. Consider revisiting.

We agree with your suggestion; the revised Abstract reads, “However, because of its mechanism of action, andexanet requires a cumbersome dosing schedule, and its use is associated with the risk of thrombosis.” Please see line 28-30.

3. “novel specific reversal variants that exhibit clotting”. The phrasing “exhibit clotting” is clinically inaccurate. Clinicians/hematologists often refer to “clotting” in terms of pathological thrombosis (e.g., DVT, PE). Here I believe you are referring to the anticoagulation reversal activity of your novel specific

variants. The way it is currently written, I am unsure whether you are referring to in vitro reversal of FXa anticoagulant effect or hemostasis (i.e., cessation of bleeding) in an animal model. Consider revisiting.

As suggested, we have added clarity and have revised the statement, “*exhibit anticoagulation reversal activity in the presence of FXaDOACs.*” Please see line 31-32.

4. “effective at low doses and do not inhibit TFPI, and consequently, are less likely to be thrombogenic”. It’s not the dose alone (e.g., mg) that would be relevant in terms of thrombogenicity. Affinity, mechanism, etc. would all affect thrombogenicity. If you are comparing to andexanet alfa, maybe molar equivalent, or binding affinity, would be a better way to compare relative amounts required to achieve a reversal effect.

We agree with the point you are making and have revised the statement to read, “*Importantly, the FXaDOACs reversal agents we designed, unlike andexanet, do not inhibit TFPI, and consequently, may have a safer thrombogenic profile*” Please see lines 34-35.

5. Throughout the abstract you refer to a “specific agent” without naming the agent. Although this is implied after reading the entire article, this will be confusing to readers who view the abstract alone. If you are referring to andexanet alfa, then simply say andexanet alfa.

Throughout the revised Abstract, we explicitly refer to andexanet alfa rather than implying it.

Introduction –

1. Typo - forecasted to be the fourth highest seller among all pharmaceuticals. Please revisit.

Thank you for identifying this typo which has been corrected. Please see line 64.

Results –

1. “Using NMR” – Please define acronyms with first use; if used only once, please spell out.

We have spelled the acronym out as it is not used again, we have not provided the abbreviation. Please see line 108.

2. Same for RSMD, please define at first use.

We have defined the acronym. Please see line 141.

3. “The increase in the volume of the apixaban binding pocket also possibly translates to loss of contacts with P4 apixaban moiety and consequently lower sensitivity towards FXa.” Please revisit, do you mean “and consequently lower sensitivity towards Fxa inhibitors”?

Thank you for identifying this error which has been corrected. Please see line 138.

4. “The low concentration corresponds to the recommended clinical dose.” What clinical dose are you referring to (e.g., rivaroxaban 20 mg once daily, rivaroxaban 15 mg twice daily?). Please provide a reference to support this statement.

We have clarified what dose we are referring to and included the reference. Please see lines 166-167 and reference 27 in the revised manuscript.

5. RDR2_2 and HI_8 activity in apixaban and rivaroxaban-treated plasma – Please refer to the fact that you are assessing activity using a thrombin generation assay. This was unclear to me on first readthrough until I saw the title of Figure 3.

We have included the TG assay in the subtitle which in the revised manuscript reads, “*RDR2_2 and HI_8 activity measured using the Thrombin Generation Assay in apixaban and rivaroxaban-treated plasma*”. Please see line 162.

6. “Addition of both low (Figs. 3A & C) and high (Figs. 3B & D) concentrations of apixaban (Figs. 163 3A & B) or rivaroxaban (Figs. 3C & D) result in loss of FXa activity.” Are the figure labels you cite in this sentence correct? You state figures C and D are for apixaban, but in the Figure 3 legend, it says “apixaban (A, B) and rivaroxaban (C, D) treated plasma”. Please check. Consider adding descriptive labels for each letter A, B, C, D to make it easier to identify the data you are presenting.

We thank you for identifying this error. In the revised manuscript we have corrected the legend for Figure 3. Please see lines 179-180.

7. Figure 3. In the introduction you propose that these Fxa variants are less thrombogenic than andexanet alfa due to lack of interaction/binding to TFPI. Your thrombin generation data seems to suggest an overcorrection of thrombin peak height (Figure 3A and C) at the low (clinically relevant) FXa concentrations (100 nM). Could the RDR2_2 variant be leading to a hypercoagulable effect through a different mechanism?

We agree with you. We revised our manuscript (please see lines 35, 213-215 and 299-306) to clearly indicate the limitations of our safety data with respect to thrombogenic risk. Our variants, unlike andexanet alfa, do not inhibit TFPI and that this may translate into reduction of TFPI-mediated thrombogenic risk to patients. Nonetheless, procoagulant activity of the variants we designed deserves further investigation because these molecules have proteolytic activity which andexanet alfa lacks. This potential procoagulant activity is evident as overcorrection of TG. We agree with you that the potential thrombogenic mechanism(s) differs from that of andexanet alfa because the variants we designed do not exhibit interactions with TFPI. The mechanism of this observed hypercoagulability is probably associated with the active enzyme being used to correct FXa inhibition. While acknowledging these limitations/caveats, it is nonetheless important to note that the observed overcorrection occurs at concentrations that are >10-times higher than the effective dose of the FXa variants. The effective dose being defined at the concentration that is sufficient to restore hemostasis to normal level upon anti-FXa treatment.

We again thank you for raising concerns about potential hypercoagulability which should be communicated to the reader. This has been done at multiple places in the revised manuscript: Please see lines 35, 213-215 and 299-306.

8. “Thus, the Fxa variants designed by us to circumvent DOAC associated bleeding, are unlikely to be thrombogenic.” Unlikely to inhibit TFPI. You cannot conclude the variants you have generated are unlikely to be thrombogenic based on this mechanism alone. Your TGA data shows evidence of hypercoagulability based on overcorrection of peak height, and in fact, the overcorrection parallels results obtained for andexanet alfa for the low (100 nM) Fxa concentrations (Figures 3A and C).

We agree with your comment (please see our detailed response to Comment #7 above) and the need to discuss the potential thrombogenicity risks at high concentrations of the FXa variants. This has been done in the revised manuscript. Please see lines 299-306.

9. “This data suggests that administration of FX in the activated form (Fxa) may be necessary for the reversal of DOAC associated bleeding.” What do you think the clinical implications off this might be? Could the administration off Fxa lead to potential thrombotic risk? Could this be a potential explanation for the overcorrection of peak height seen in Figures 3A and C? Wouldn’t the administration of a zymogen be desirable since the bypassing protein would be regulated by endogenous coagulation processes? (e.g., VMX-C001 is a zymogen).

Based on existing data we agree that the clinical risk of thrombosis should be considered and discussed (see responses to Comment #s 7 & 8 and associated changes in the text). You raise a valid question of potential thrombogenic risk caused by administration of the variants as active enzyme, and the observed overcorrection of TPH. We again emphasize that the mentioned overcorrection is observed at the concentrations of variants that far exceed the concentration needed for reversal of FXa inhibitors. Nonetheless, should this work progress to a clinical stage, it would be important to consider using our variants in a zymogen form. To date, we have only investigated the zymogen form in functional tests.

We would also like you to consider a plausible explanation of the observed overcorrection. As seen in the figure below (Fig. 1), the TG assay shows loss of sensitivity at high concentrations of FXa variants. Starting from a FXa concentration of $\sim 6 \mu\text{g/mL}$ we can observe the absence of lag time and presence of intensive clotting right after the reaction start for variant RDR2_2, and the cleavage of substrate by variant HI_8 before production/peak thrombin:

Fig. 1

We also cannot exclude the fact that variant RDR2_2 also cleaves fluorogenic substrate at high concentrations, and we cannot separate these 2 processes, i.e., cleavage of substrate by the variant vs cleavage of substrate by produced thrombin. Thus, at high FXa concentrations, we cannot determine whether the observed hypercoagulation is due to thrombin production, or an artifact caused by substrate cleavage.

Moreover, when we directly compare the variant RDR2_2 with wild-type FXa (Fig. 2) we observe that the variant RDR2_2 has a lower signal at high concentrations (with undetectable peak for wtFXa due to

intensive clotting). This implies that the potential thrombogenicity of RDR2_2 is at least less than that of wtFXa:

Fig. 2

Discussion -

1. Typo – “Two products currently used to control bleeding associated Fxa-targeting DOACs”. Please check.

We corrected the sentence. Please see lines 228-229.

2. “their use is associated with significant risks of thrombotic adverse events”. This statement is too vague. There is still debate (since it is hard to prove) that thromboembolic events post reversal are directly attributable to the reversal agent in question. Some of these thromboembolic events might just be due to the underlying indication for anticoagulation (e.g., atrial fibrillation and risk of stroke). I would consider rephrasing this and simply cite the 30-day risks of thromboembolic events observed post hemostatic therapy and mention that some of this risk might be attributable to the hemostatic therapy itself (e.g., TFPI interaction for andexanet alfa; administration of supraphysiological levels of prothrombin for PCCs). You do state “The increased risks of thrombosis may be due to inhibition of TFPI in the case of andexanet alfa and an increase in plasma levels of prothrombin, FX and FIX following use of PCCs.” later in the paragraph but could also consider adding that some of these events may be due to underlying risk rather than the anticoagulation reversal itself.

We agree you that this statement might be too vague and have modified the language. Please see lines 233-235.

3. “particularly when compared to an alternate reversal agent that includes sequences derived from the venom of the elapid snake (*Pseudonaja textilis*).”, a reference would be needed here, which reversal agent are you referring to?

We decided to remove the statement about an alternate reversal agent that includes sequences derived from the venom of the elapid snake (*Pseudonaja textilis*) because there is no evidence about how immunogenic this molecule may potentially be (too speculative at this moment); moreover, this agent is

currently being investigated in early phase clinical trials and not yet approved for human use for the reversal of FXa inhibitors like apixaban or rivaroxaban.

Methods –

1. Re: thrombin generation assay, did you consider using a calibrated assay (calibrated to thrombin units, rather than RFU). We're any other parameters collected (e.g., total thrombin generated; analogous to e.g., endogenous thrombin potential in the case of Calibrated Automated Thromography). If trying to estimate hypercoagulability, area under the thrombin generation curve might be a more appropriate measure than peak height. If both were measured, both should be reported, along with kinetic parameters (lag time and time to peak).

Thank you for this comment and for the attention you have given to the details of the assays we are using. It is true that the TGA data is usually shown in calibrated format, as commercial TG assays permit automated calibration, and the data is automatically provided in nM of thrombin. We are using an in-house assay with the conditions optimized for the purpose of evaluating the FXa variants in the TG test. Please note that while we can convert the RFU to nanomoles of thrombin the absolute values would still deviate from those obtained in a standardized CAT test. Importantly, the purpose of our optimized assay was to show the difference between variants designed in this study and andexanet alfa, and the ability of the FXa variants to reverse the effect of FXa inhibitors. Thus, we are more interested in comparing the different proteins/variants and absolute quantitative values are less important. We feel that it may be misleading to present our data in nMs because the values presented may not be identical to those reported using an off-the-shelf CAT assay. We are proving below for your evaluation the data shown in Fig. 3 of the revised manuscript, presented as nM Thrombin rather than RFU. As shown in Fig. 3 (in this document), such a conversion does not change the overall conclusions.

Fig. 3

We consider Thrombin Peak Height (TPH) as the most representative and responsive parameter of TGA as it clearly shows the difference between FXa variants and their effects in plasma. Time parameters (time to peak (TTP) and lag time (Tlag)), as well as ETP, appeared to be less representable, not very robust (for example, Normal Pooled Plasma's ETP range is so broad that no hypercoagulability would be detected). Also note that in the presence of high concentration of anticoagulants (such as DOACs) there is no coagulation and therefore no values will be obtained for the time parameters (e.g., TTP, Tlag). Keeping in mind the limitations of each of these parameters we are providing the curves for your evaluation. Please see Figs. 4-6 in this document. Our conclusions remain unaffected by using parameters other than TPH.

Fig. 4

Fig. 5

Fig. 6

Given the limitations of alternate parameters, our preference is to present our TPH data as RFUs/min in the text.

Reviewer #2 (Remarks to the Author):

The authors have engineered blood coagulation FX to produce antidote molecules that could be used by patients that receive direct oral anticoagulant molecules (DOACs) targeting factor Xa, such as apixaban and that would be at risk of bleeding, due to, for instance, a car accident or related threatening conditions. There is indeed a need for such “antidote” molecules as more people are likely to take anticoagulant therapies (e.g., ageing of the population). The available agent that reverses the effects of FXa DOACs named andexanet alfa (catalytically inactive truncated variant of human FXa) has numerous adverse effects and drawbacks.

The study is therefore of high interest. Below are my questions/comments following the order of the article.

A) The authors wrote: The inhibitor binding site of FXa is located on the S1 and S4 sub-pockets of the catalytic domain

This tends to be confusing and could be optimised. The catalytic site of all serine proteases contain different subpockets, all pockets can contribute to the free energy of binding, some, in addition, are important for specificity/selectivity.

We thank the reviewer for their suggestion. The revised text reads, “*The DOACs bind to the active site of FXa and blocks the access for its substrate. Structural analysis of FXa in complex with apixaban shown that it interacts with amino acids located on subsites S1 and S4 of the active site.*” Please see lines 72-74. Please also consider that subsequent lines (75-80) also add additional details of the molecular interactions between apixaban and FIXa.

B) The authors want to design FXa variants that are catalytically active in the presence of the DOAC apixaban. They use a Rosetta protocol for this purpose, with two main goals:

B1) Identified mutations that would adversely affect FXa function or not permit efficient expression of the protein

This is unclear, to the best of my knowledge, Rosetta cannot tell if a mutation can lead to efficient expression of a protein? What seems to be done here is to probe if mutations around the active site would affect the stability of the protein? So, it should be something like a delta-deltaG? The authors should explain in the supplement what is being computed and explain the Rosetta method used. The readers do not know if the provided scripts are commonly used in the field or if they carry something unique.

Thank you for raising these important distinctions which will clarify matters for readers. You are correct, Rosetta does not provide information as to whether a mutation can lead to efficient expression of a protein, and the text has been modified accordingly. We have made changes to the manuscript (in the main text as well as the supplement) to make the design strategies clear to the reader (Please see lines 83-96, 317-338 and added additional explanations in the Supplementary file under “Computational scripts and files”). We were mainly concerned with the mutations around the active site could be beneficial (in filling in the void) when apixaban was not present and we used total_score as our metric to determine if they were deleterious mutations or not. Then total_score of the input structure and the total_score of the designed models were used to determine top-scoring and to determine which mutations to look at first (the change in total_score).

B2) The authors also want mutations that decrease the affinity for apixaban. Here it is unclear, the authors talk about top-scoring designs (with respect to binding site contact modifications)? Please translate. A predicted binding score? Creation of cavities/clashes among amino acids in this region and thus potential local destabilisation? What is being monitored? And then the authors investigate manually potential clashes between the mutant FXa and apixaban? Why not looking at binding scores directly? What if it is not a clash but modified electrostatic interactions? All this section is highly unclear and should be explained in the supplement.

Thank you for identifying the lack of clarity in our explanations. We trust the extensive rewriting of the manuscript detailed in response to Comment B1 address the issues you have raised. Briefly, we looked at the change in total_score and cavity_score that helped us to identify the residues that would not affect protein stability but disrupt FXa-apixaban interactions.

We did not look at a direct “binding score” metric since we were interested in favorable mutations in the substrate binding site without the apixaban. The Rosetta design process was performed without apixaban occupying substrate binding pocket.

B3) then the authors use the DALI server to identify other serine proteases from the coagulation that could give some insights to introduce mutations.

The reason for DALI is not clear here. It is well known that serine proteases from the coagulation are relatively similar, at least in the serine protease domain, with some key differences in amino acids here and there and in insertion/deletion of some loops. What would seem important, in addition of DALI or instead of DALI, is to have a multiple sequence and structural alignment that will show at the amino acid level, what is conserved, where we have insertion or deletion in the family so as to put a score on the importance of selecting some regions for mutations/grafting, to see which amino acids are risky to mutate while also showing the secondary structure elements of FXa... Such a multiple sequence alignment in the supplement would be welcome and is easy to do.

We agree with you that protein design can be approached in different ways. However, the more information obtained during the design process, the better. Our rationale for selecting the DALI approach was that in some cases, structure comparison can reveal distant evolutionary relationships that may not be identified in sequence alignments. We agree that keeping evolutionary conserved amino acids is important, and we therefore compared Factor X sequence with other vitamin K dependent serine proteases including Factor VII, Factor IX, prothrombin, and protein C. The results revealed W215 to be conserved in the protease domain. In the literature, this residue has been demonstrated to be a key player in regulating the accessibility of substrate to the active site of the enzyme (<https://onlinelibrary.wiley.com/doi/full/10.1111/jth.15897>). Additionally, we decided to evaluate what is the activity trade-off when W215 is mutated. It turned out that this variant (W215A) exhibits sufficient activity to reverse the inhibition of FXa activity in the presence of 2µM apixaban. As for the STYVPG inset, the initial sequence alignment identified protein C and FVII as two potential candidates for the loop replacement due to the same length of the loop; further analysis using Dali server identified FVII as a top candidate due to the higher DALI score, and therefore, we used this sequence.

The authors wrote that they measure apixaban binding to FVII using NMR. It is surprising that such data are not available for apixaban and most coagulation proteases (at least) as such information would be mandatory during the drug development process. As such, just taking this information would help to select some proteases from which to get some insights in term of mutation, and then go to the 3D.

This information is publicly available (<https://www.ncbi.nlm.nih.gov/pmc/articles/PMC3090580/>). However, given that this is a critical parameter for our design process we selected to also execute the experiment ourselves in-house to confirm, using proteins that were expressed and synthesized by us that apixaban does not bind to FVII.

The selection of FVII STYVPG to replace a related loop in FX should be shown in the supplement (the loop has the same size... which could then give some insights about potential immune reactions...).

Per your request we have included the alignment of the vitamin K dependent serine proteases in the supplementary information as Figure 1:

	88	95	96		108	210	215	221
FX	V I K H N R F T - - K E T Y D F D I A V L R L ...					T G I V S W G E G C A R		
FIX	I I P H H N Y N A A I N K Y N H D I A L L E L ...					T G I I S W G E E C A M		
Prothrombin	I Y I H P R Y N W - R E N L D R D I A L M K L ...					M G I V S W G E G C D R		
FVII	V I I P S T Y V - - P G T T N H D I A L L R L ...					T G I V S W G Q G C A T		
Protein C	V F V H P N Y S - - K S T T D N D I A L L H L ...					V G L V S W G E G C G L		

What is known about the introduced mutations in FX? Are they present in patients?

You raise an interesting point, having checked the available literature we found that none of the mutations we introduced have been identified in Factor X deficient individuals.

<https://www.sciencedirect.com/science/article/pii/S0268960X21000394?via%3Dihub>

<https://www.thieme-connect.com/products/ejournals/html/10.1055/s-0029-1225763>

C) Then the authors evaluate the procoagulant potential of the two selected FX variants. They wrote: The reduced activity can be explained by the destabilization of the pocket This is unclear, the pocket is not destabilised? The fitting of the inhibitor is perturbed, the pocket might become more flexible or more rigid... but destabilisation here would mean some kind of deltadeltatG issue, and potentially folding issue.

We agree with you that the language was unclear and have changed “destabilization” to “disruption”. Please see line 134.

D) in the abstract the authors talk about andexanet alfa, in the discussion they add PCCs, thus the two strategies should be mentioned in the introduction?

In the introduction (lines 48-51) we state, “Recent meta-analysis of reversal agents demonstrated that effective hemostasis rate using andexanet alfa is comparable with Prothrombin Complex Concentrates (PCCs) that are used for the nonspecific reversal of DOACs”.

E) This sentence should be clarified: The design process included: (i) Design of an ‘empty’ apixaban binding pocket. What is an empty pocket?

Thank you for your suggestion. We have clarified the sentence by changing the “empty apixaban binding pocket” to “substrate binding site which is also the DOACs binding pocket” (please see line 239).

F) The computer strategy is somewhat surprising as very simple alternative could lead to the same

results (e.g., Multiple sequence alignment and literature search).

In conclusion, the reported experimental data seem scientifically sound but several clarifications are missing with regard to the computational approaches. For instance, why the authors used fixed backbone Rosetta energy minimization while it is known that proteins are flexible and that small adjustments in the backbone are usually needed after introduction of amino acid changes? So again the computational approaches need to be explained further, if there is no room in the main paper, this can be done in the supplement. The way MD is used here is surprising, free energy of binding with the ligands could be computed for the wt and mutants instead or in addition of monitoring the pocket volume?

Regarding the fixed backbone work, we clarified that fixed backbone (applying atom coordinate constraints) was only used during the FastRelax protocol (no design) to allow for sidechain-rotamer optimization of an input structure, which is standard practice before performing any design. We have also provided the XML files, in the supplementary file, used for FastRelax and design that show whether constraints were applied or not. The atoms coordinate constraints were removed during the sequence design process to allow for more movement and allow Rosetta to fill in the gaps left by the removal of apixaban. This is shown in the removal of the "<" in the Add mover="poseCST"/> located in the <PROTOCOLS> section of DesignSimpleFRFX_noligand.xml (please see Supplementary file under "Computational scripts and files", p8)

We agree with you that free energy calculations of the ligand with WT & mutants can be computed. However, for our purposes Using GROMACS and FPocket was sufficient to investigate the stability of the binding pocket to support our hypothesis and show the instability of the binding pocket which would impede binding between the protein and apixaban.

REVIEWERS' COMMENTS

Reviewer #1 (Remarks to the Author):

Thank you. My comments have been addressed. I congratulate the authors again on this great work.

I understand and accept the rationale as to why the authors would prefer to only present peak thrombin generation for this current paper/analysis.

For future research/submissions on this topic (not for the current revision), I would recommend the authors endeavor to measure and report all TGA parameters (lag time, TTP, ETP, in addition to peak), since direct Xa inhibitors affect not only quantitative parameters of thrombin generation (peak, ETP), but kinetic parameters as well (prolonged lag time).

Reviewer #2 (Remarks to the Author):

I would like to thank the authors for clarifying several points in the manuscript. From my side I believe that this revised version is ready for publication.